

# Design of dynamic ski motion capture and human posture detection model based on spatio-temporal Transformer and wireless device tracking technology

Xiaohua Wu[1], Jian Yi[2], Yongtao Shi[1] and Gong Zhang[1]

[1] Hebei Sports University, Shijiazhuang, China
[2] Capital Normal University, Beijing, China

## ABSTRACT

As a result of significant advancements in living conditions, individuals have redirected their attention towards physical exercise. Skiing, as a widely popular sport, necessitates the real-time maintenance of correct posture during movement. Therefore, we present a dynamic skiing motion capture and human posture detection model that leverages wireless device tracking. Primarily, personnel tracking is enabled through the construction of service base stations and the utilization of wireless device tracking technology. Subsequently, a human posture detection model is formulated in the form of human posture key points, employing the image information of each frame obtained via wireless devices. Finally, we introduce a spatio-temporal Transformer structure that facilitates the detection and recognition of human posture in consecutive frames. Our results demonstrate that our approach can precisely locate and track the position of skiing personnel. Compared to the latest Blip and Conformer methods, our technique yields $F$ values that surpass them by 1.20% and 4.51%, respectively. Moreover, our model can achieve convergent model parameters and accomplish training objectives more efficiently, thus enabling posture detection and dynamic capture of skiing personnel *via* image and video information.

## INTRODUCTION

With the aid of advanced technology, individuals have become increasingly health-conscious, leading to a rise in outdoor sports participation. Among these activities, skiing holds a special place in the hearts of many due to its challenging and exhilarating nature. Skiing serves as an excellent test of human coordination and posture balance. Along with ensuring an adequate amount of exercise, real-time monitoring of the human body's posture during exercise is essential to guarantee life safety. As such, we have directed our focus towards motion capture of skiing and human posture, and conducted research on safe skiing using wireless device tracking.

Wireless device tracking is a widely used technology in sports, with a primary focus on two aspects: sports monitoring and auxiliary training. It enables real-time tracking of

Corresponding author
Jian Yi, yijian5436@126.com

people's motion data, including position, speed, acceleration, and attitude, allowing for analysis and monitoring of the motion state. This enables individuals to better understand their motion state and optimize their motion plan (*Seifeldin, El-Keyi & Youssef, 2011*; *Zeng et al., 2022*; *Xingyu, Lijun & Jiaquan, 2022*). Moreover, wireless device tracking can track people's physiological indicators such as heart rate, respiration, and more during exercise. By combining motion data analysis, it can aid in the prevention of sports injuries and promote better health. In addition, incorporating virtual reality technology, wireless device tracking can facilitate intelligent auxiliary training, providing the public with a more intuitive and enriched sports experience and feedback.

By utilizing wireless device tracking technology, it has become viable to dynamically capture and detect skiers' posture in real-time. Employing a plethora of technical methods such as sensors, cameras, computer vision, and deep learning models, the dynamic capture of skiing performance can effectively discern and evaluate critical indicators, such as posture, position, speed, acceleration, and other pertinent metrics during the skiing process. Consequently, this facilitates the monitoring, analysis, and optimization of skiing athletes' motion states and performances. With dynamic capture, the movement state and performance of skiers can be comprehended effectively, and valuable data feedback can be provided to coaches to improve athletes' training plans and elevate their performance levels (*Yi et al., 2022*). Real-time human posture detection refers to the process of detecting and analyzing the human body in video streams to acquire position information of crucial points of the human body under various postures, such as the coordinates of the head, shoulder, elbow, knee, and other crucial points. Through this, the posture detection and analysis of the human body can be accomplished. Typically, real-time human posture detection is implemented by means of deep learning models. With the assistance of training data, these models can rapidly and accurately identify and track the crucial points of the human body in real-time video streams, thus enabling the real-time detection and analysis of human posture. Real-time human pose detection can be applied in various fields, including sports training, health management, and virtual reality, to offer people a more immersive, intuitive, and data-driven interactive experience (*Bressel, Smith & Nash, 2022*). Through the wireless collection of skiers' information during skiing, we can obtain their sports status and human posture in real-time (*Li et al., 2022b*).

Therefore, this article aims to ensure the safety of individuals while skiing, guide beginners to adopt correct skiing postures, and monitor people's skiing status in real-time. To this end, we propose a dynamic skiing motion capture and human posture detection model based on wireless device tracking, which serves to expand the information data of skiing and ensure skiing safety. In order to highlight the human posture in the image, we propose a structure of body key node and propose a feature attention method based on network fusion. Further, we propose a spatio-temporal Transformer structure, which is suitable for ski sports and integrates the detection module of key nodes described in the former.

The main contributions are as follows:

1. We propose a construction method of tracking base for people based on wireless to help mangers locate the position of skiers.

2. We extract spatial and semantic features of the image to enhance the region of skiers, which can help model understand the pose.

3. We propose a method to construct spatio-temporal for Transformer for the dynamic capture of skiing motion.

# RELATED WORKS

## Computer vision-based human motion capture and human pose detection

Motion capture systems based on machine vision utilize image processing to extract marker-free human motion information from images of the human body. In computer vision, the main goal of human motion capture is to extract spatial feature information from a single frame image or video sequence taken from one or multiple unsynchronized or synchronized camera views, and subsequently recover or record information related to human motion and posture.

Image-based tracking is one of the most commonly used methods for visual human pose capture. Optical tracking methods are typically employed with the help of a camera and computer. A camera captures a video image of human motion, and computer algorithms then analyze the motion trajectory of the main joint points of the human body using image recognition algorithms, resulting in the recognition of the human posture. *Toshev & Szegedy (2014)* proposed the DeepPose model, which formulates the pose estimation problem as a human joint point regression task and uses a deep convolutional neural network for global inference. *Ouyang, Xiao & Wang (2014)* replaced support vector machines with deep neural networks and performed nonlinear inference on the body part information calculated by the hybrid part model, resulting in superior results. *Chu et al., (2016)* incorporated prior knowledge about the spatial relative positions between human joints to introduce geometric transformation kernels and a bidirectional tree model, which enabled the training of a convolutional neural network to learn the dependencies between human joints, thereby improving human pose estimation results. *Ju & Kml (2018)* employed a 2D human estimation method based on a convolutional neural network to obtain the 2D joint point heat map of the target human body. They then trained a 2D to 3D lifting network to obtain the 3D estimation results of the target human body. *Tome, Russell & Agapito (2017)* proposed a joint task framework for 2D human joint point estimation and 3D human motion reconstruction that combines the prior knowledge of pose in 3D using a multi-stage convolutional neural network model. *Pavlakos et al. (2017)* created a fine discrete 3D space surrounding the target human body and trained a convolutional neural network to learn the likelihood that each target human joint point is located at a specific 3D space point.

## Wireless device tracking technology

The methodology utilized to estimate the location of a target wireless device is founded on measuring the signal strength amidst neighboring wireless devices. This enables the tracking of the device, although the accuracy of this method is hindered by environmental interference that can hamper the propagation of the wireless signal. To tackle the issue

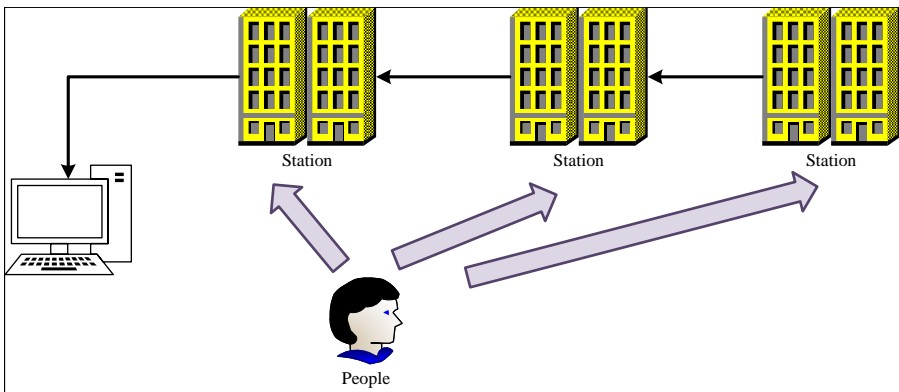

**Figure 1** **The policy of wireless device tracking.**

of time-varying environments, *Wang et al. (2012)* proposed a resilient scheme that leans on the difference method. Since environmental changes usually transpire gradually, the impact of measurements taken at adjacent moments is often negligible. By computing the difference of adjacent moments, it is feasible to determine whether a link is obscured by the target, thereby alleviating some of the challenges associated with time-varying interference such as temperature and humidity. To address the issue of burst noise, *Givehchian et al. (2022)* presented a solution. The surging popularity of wireless networks has led to congestion in the limited frequency band, creating difficulties in the positioning and tracking of devices. *Kaltiokallio, Bocca & Patwari (2012)* have put forth a method to overcome this problem of device channel selection. They have illustrated that permitting the device to operate in a channel with high interference can impede the accuracy of device tracking. Therefore, they have proposed evaluating the quality of each channel in the offline state, selecting the channel with less interference, and having the device operate on this channel in the online phase (*Mobsite et al., 2023*; *Mei et al., 2023*; *Liu et al., 2022*; *Ma et al., 2022*). While this approach mitigates co-channel interference to some extent, there still exists the possibility of interference transpiring in the device's current operating channel during online operation (*Gulati et al., 2009*; *Ye et al., 2021*).

## DYNAMIC SKIING MOTION CAPTURE AND HUMAN POSTURE DETECTION MODEL BASED ON WIRELESS DEVICE TRACKING

To accurately track the movement of skiers, we present a scheme for constructing a personnel tracking base station based on wireless devices, as depicted in Fig. 1.

In the ski scene, multi-directional ranging positioning is formed by constructing multiple base stations and a remote-control terminal. The skier simultaneously calculates the distance between it and multiple base stations, and according to the principle of wireless device tracking, the position of the skier can be directly calculated. Then, the remote-control terminal is used to point the camera in the scene at the skier, which provides the hardware and network basis for the subsequent detection and analysis of ski attitude. The principle of

wireless equipment tracking involves receiving the radio signal emitted by the equipment and employing certain algorithms to compute its position.

Considering the operating cost and construction cost, we built multiple signal base stations to realize the location of skiers. First, we determine the rough location of the person by receiving signals from at least three base stations or receivers, where the base station or receiver measures the arrival time of the device's signal with high precision clock and hardware support. Then, we use the strength of the signal from the receiving device to pinpoint the location of the device based on RSSI technology. Signal strength is usually related to the distance between the device and the receiver. By collecting signal strength from different locations, a model of signal strength can be built to estimate the location of the device. Finally, we can further pinpoint the skier's location by measuring the time it takes the device's signal to reach different base stations or receivers.

By employing the above three methods, we can effortlessly track the dynamic position of skiing personnel, thereby enabling the mobilization of video sensors for real-time monitoring of skiing personnel to ensure their personal safety.

## Human posture detection model

The ski attitude detection is different from the ordinary image detection task. The detection of ski attitude has very high similarity within class, and the detection area is irregular. Considering the above reasons, we quantified the human posture into several fixed nodes. In order to highlight the key points of the human body, we need to enhance the features of the image. To achieve the precise capture of skiing movements, detecting human posture in the image is of paramount importance, as illustrated in Fig. 2, in order to assess the skier's form during their performance. Additionally, we demonstrate the human body's key points in Fig. 3.

To begin with, we employ the Mosica data augmentation network to augment the input image data using random scaling, cropping, and arrangement. We should explain that the use of enhancements here is to enrich the information in an image. By fusing the information in the four graphs into one, the detection of small targets can be realized, which is exactly suitable for the characteristics of the key nodes in human body. By enriching the dataset, the network's training speed is greatly improved, while also reducing the model's memory requirements. The Focus structure is then utilized to perform feature transformation on the input image, resulting in a 32-bit feature map that is passed to the backbone network.

The Focus module splits the image into multiple pieces before entering the backbone, similar to adjacent pooling in an image, resulting in four images of similar length with no information loss. The feature channel is then extended by four times, and the resulting new image is subjected to convolution.

The underlying network architecture comprises of CSPNet and FPN structures, which extract features from both spatial and semantic information of the image. With CSPNet and FPN, we can enhance the area of people's key nodes and make them more prominent in the image. Meanwhile, we can solve the problem of different dimensions and coincidence of key nodes in the image. In addition, FPN and CSPNet have stability for different sizes of poses

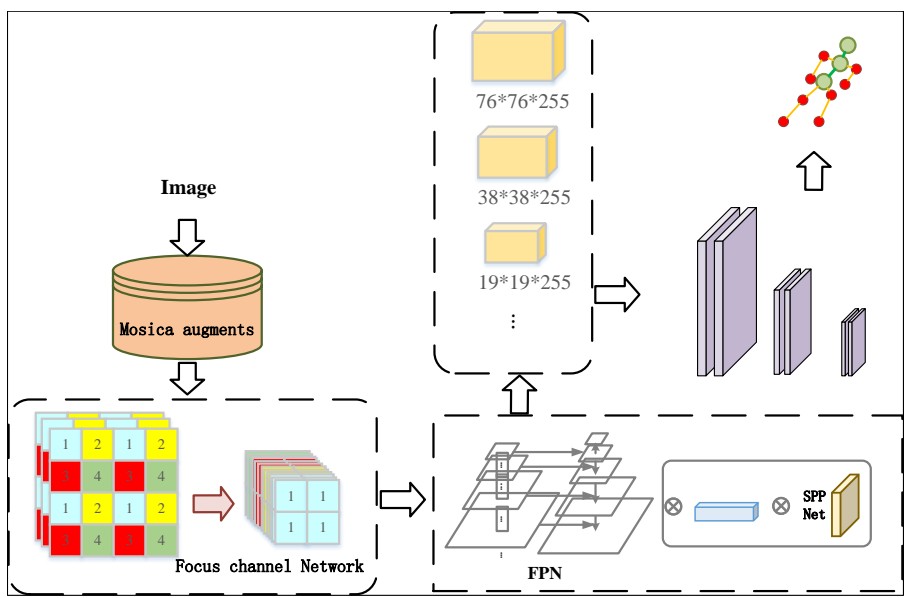

**Figure 2** Overview of our human posture detection model.

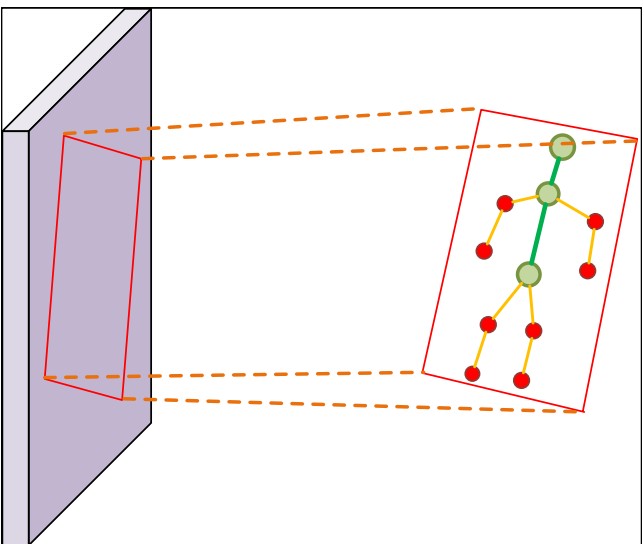

**Figure 3** The settings of human posture detection.

in the image. This leads to the generation of three-dimensional feature layers that facilitate candidate box prediction, enabling the detection of human posture and obtaining key point images of the human body. Upon obtaining the key point posture image, we devised an excellent posture classification algorithm based on the VGG16 network. Following image transformation, the input image undergoes 13 convolutional layers and three fully connected layers. The first two 64 × 64 convolutions are succeeded by two 128 × 128

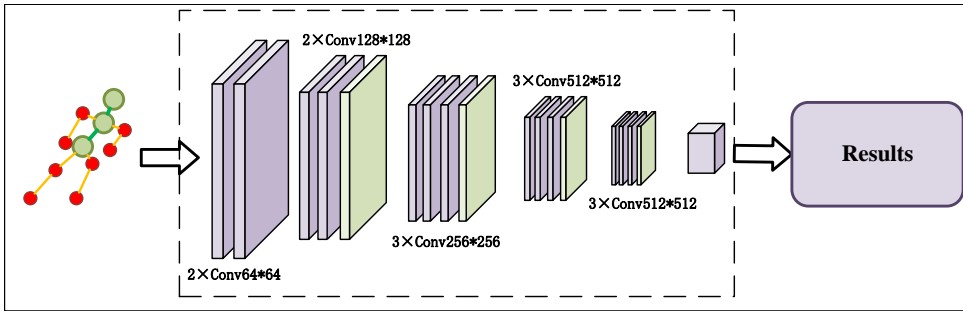

**Figure 4** Ski body posture classification structure.

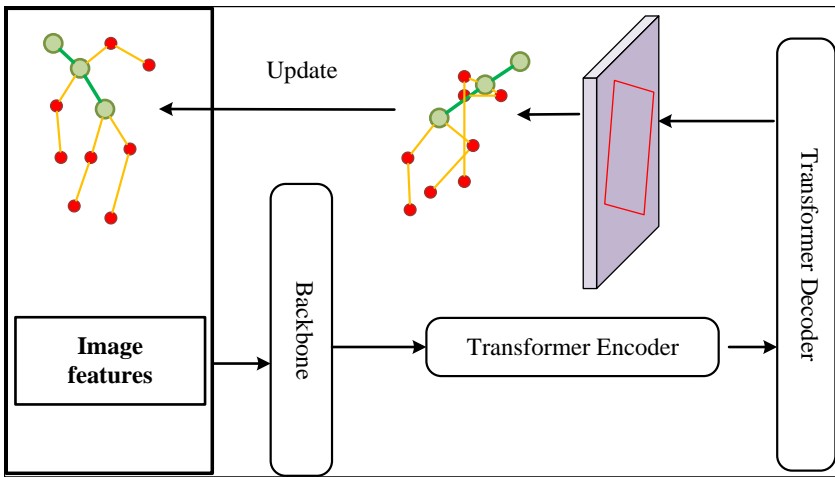

**Figure 5** Ski dynamic capture method based on time-space Transformer.

convolutions, followed by three 256 × 256 convolutions and two 512 convolutions, each module concluded by a pooling operation. Finally, after three cycles of full connection and softmax activation function, the feature classification results for the input image are generated, leading to the realization of excellent skiing human posture judgment. Refer to Fig. 4 for an illustrative overview of the process.

## Ski motion dynamic capture method

By analyzing a single frame image, we can classify a human's skiing posture, evaluating its strengths and weaknesses at that particular moment. To capture the real-time dynamics of skiing motion, we propose a method for dynamic capture of skiing motion based on spatio-temporal Transformer, as illustrated in Fig. 5. Considering that the object of our study is continuous video frames collected by wireless tracking equipment, we propose this method. In addition, we combine the characteristics of ski sports in the model and embed

the space–time mechanism for the detection of key nodes for human body to realize the continuous monitoring of ski sports.

In addition, we need to explain why we don't use LSTM and Transformer. LSTM and Transformer are targeted at serialized tasks and the dynamic capture of skiing exactly has a certain sequence. However, motion capture is a long sequence task. LSTM and Transformer are not usually used to handle long sequence tasks due to hardware constraints. In order to satisfy the property of long sequence and have a certain accumulation memory forgetting mechanism, we propose a spatio-temporal model suitable for skiing dynamic capture.

The fundamental structure of spatio-temporal Transformer is that of multi-head self-attention, wherein the key component is self-attention. The principle of this component is demonstrated in the following formula:

$$\hat{z}_i = MSA(LN(z_{i-1})) + z_{i-1} \tag{1}$$

$$z_i = MLP(LN(\hat{z}_i)) + \hat{z}_i \tag{2}$$

$$Attention(Q, K, V) = softmax\left(\frac{QK^T}{\sqrt{D}} + B\right)V \tag{3}$$

where Q, K and V denote the Query, Key and Value matrices, respectively. The LN is the LayerNormlization and MLP refers to Multilayer Perceptron. Besides, $z$ presents the image features.

This article adopts a direct classification and regression method for feature vector analysis, augmented with the confidence branch update template to enhance the algorithm's robustness. Since dynamic ski motion capture is a continuous object, it is necessary to discard long-term accumulated memory information for the model. Therefore, we embedded the confidence branch update template to make the model forget long-term accumulated memory information and enhance the connection between adjacent frames, and ensure the accurate capture of skiing action. The classification and regression branches comprise of a three-layer perceptron and ReLU function. The classification and regression branches predict each feature vector's vector output by the Transformer feature fusion module, obtaining foreground and background classification vectors and bounding box regression vectors.

During tracking, clipped templates become unreliable in cases of object occlusion, loss, and scale changes, so templates need not be updated under extreme interference factors. This article proposes dynamic template updates only when the search area contains targets. It adopts a parallel confidence branch as a model update strategy based on classification and regression branches. The confidence branch includes a three-layer perceptron and a Sigmoid function Transformer feature fusion module. The vector obtained by the confidence branch serves to obtain the confidence score.

If the confidence score is greater than the set threshold $\gamma$, the target in the current search area is trimmed as a dynamic template frame. Otherwise, the template does not need to

be updated. In the inference phase, the initial template and search area of the first frame are trimmed and sent to the tracking network to obtain the bounding box results and confidence scores.

$$Y = \begin{cases} \text{True if score} > \gamma \text{ and frame} = \tau \\ \text{False otherwise} \end{cases}. \tag{4}$$

when the number of running frames reaches the update interval $\tau$ and the confidence score is greater than $\gamma$, the dynamic template will be updated by the network.

## EXPERIMENT AND ANALYSIS

### Dataset and implement details

We have evaluated the effectiveness of our proposed method using the Ski 2DPose Dataset (https://www.epfl.ch/labs/cvlab/data/ski-2dpose-dataset/). The dataset consists of videos divided into 147 training sequences and 11 validation sequences of varying lengths, where each segment represents a continuous motion. We split the videos at intervals ranging from 0.3 s to 10 s because we need to consider the diversity and stability of the data to obtain more training, testing, and validation data to demonstrate the performance of our method. Through such a frame extraction method, we can ensure the uniqueness of model training, and can achieve the test results of other excellent methods on the data set. In addition, the number of each pose is determined by the difficulty. The dataset features 1,982 images of amateur to semi-professional alpine ski racers, where 24 joints, including skis and poles, were hand-annotated.

The experiments were carried out on a device equipped with an i5-12500 CPU and an RTX 3080 GPU, running on the Centos operating system, and implemented under the PyTorch framework. The model was trained for a total of 200 rounds, with a batch size of 128 and an initial learning rate of 0.001. The optimizer used for the model was SGD, with a momentum value of 0.9. and the weight decay term is set to $1 \times 10^{-4}$. When achieving the frames from the wireless device, we will never process the resolution of images. Because of the different camera resolutions, we did not change the image size when processing the image of the model. At the beginning of the model design, we take into account the differences in actual application scenarios, and the resolution of the image will not affect our deep learning model, and the accuracy will not fluctuate. In addition, we apply the Mosica data augmentation for the all methods in our article, including the comparison of other methods, to ensure the fairness in training and testing.

To evaluate the model performance, we use precision, recall, and F-measure as evaluation criteria, which are calculated as follows:

$$\text{Precision} = \frac{TP}{TP + FP} \tag{5}$$

$$\text{Recall} = \frac{TP}{TP + FN} \tag{6}$$

$$F = \frac{2\text{Precision} \cdot \text{Recall}}{\text{Precision} + \text{Recall}}. \tag{7}$$

**Table 1** performance of our method and other methods.

|  | Precision | Recall | *F* value |
|---|---|---|---|
| CNN | 72.90 | 73.69 | 73.32 |
| LSTM | 70.98 | 67.36 | 69.47 |
| CNN-LSTM | 75.40 | 76.78 | 76.89 |
| CNN-Transformer | 78.45 | 77.12 | 78.58 |
| SwinTransformer | 78.59 | 79.54 | 80.02 |
| Vit | 79.45 | 78.23 | 79.21 |
| Deit | 78.43 | 78.23 | 78.53 |
| Blip | 80.32 | 79.32 | 80.78 |
| Conformer | 78.23 | 77.49 | 77.47 |
| Ours | 80.23 | 80.90 | 81.98 |

## Results and discussion

We conducted experiments on the Ski 2DPose Dataset to evaluate the effectiveness of our dynamic ski motion capture and human pose detection model based on wireless device tracking. In addition, we also compared our method with several models that are capable of handling multimodal features, such as CNN (*Kattenborn et al., 2021*), LSTM (*Yu et al., 2019*), CNN-LSTM (*Livieris, Pintelas & Pintelas, 2020*), CNN-Transformer (*Li, Chen & Zhang, 2020*), SwinTransformer (*Liu et al., 2021*), Vit (*Dosovitskiy et al., 2020*), Deit (*Touvron et al., 2021*), Blip (*Li et al., 2022a*), and Conformer (*Samarakoon & Fung, 2023*). Table 1 summarizes the comparison results, and our method achieves the best performance, ranking first in precision, recall, and F-value with scores of 80.23%, 80.90%, and 81.98%, respectively. Compared with CNN-LSTM and CNN-Transformer, our method improves F-value by 5.09% and 3.40%, respectively. Furthermore, our method outperforms the latest Blip and Conformer method in terms of *F*-measure by 2.54% and 4.51%, respectively. Although our model has a large number of parameters and requires a relatively long inference time that can achieve the 9.77 of FPS, we consider that CNN-based models may lose the details of multimodal features, while LSTM and Transformer-based models may lose some global features and have inaccurate classification. Our model balances sensitivity and feature preservation while ensuring accuracy by appropriately cutting the structure of the model.

We present the convergence analysis of our novel dynamic ski motion capture and human posture detection model, in comparison with other established methods, during the training phase, as illustrated in Fig. 6. Our method, which is based on wireless device tracking, exhibits rapid convergence of the model parameters during training, effectively accomplishing the training objective. In addition, we showcase the human posture detection results of our approach for consecutive skiing motion frames, providing a visually appealing representation, as depicted in Fig. 7. Our method excels in dynamically capturing skiing motion and accurately reconstructing the skiing personnel's body posture, ensuring their

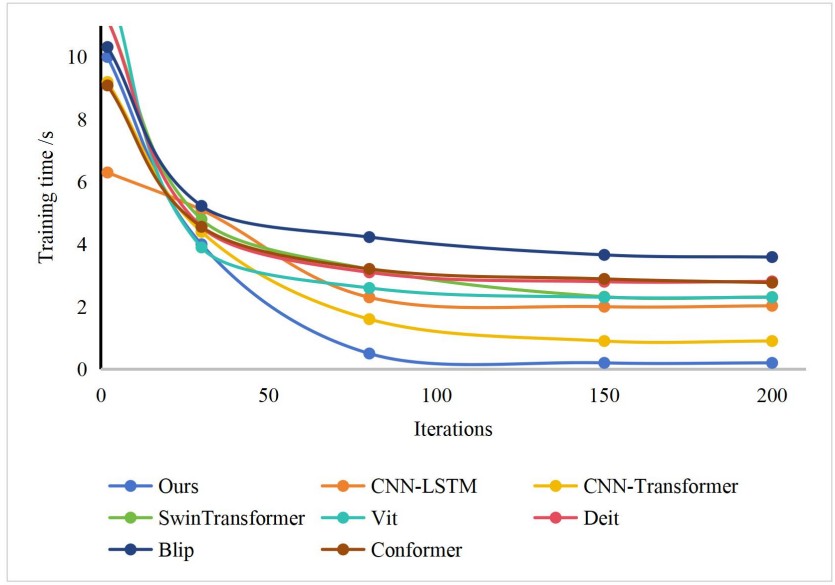

**Figure 6** The training of our method and other methods.

**Table 2** Test of our application.

| Volunteer | 1 | 2 | 3 | 4 | 5 | 6 | 7 | 8 | 9 | 10 | Our model |
|-----------|---|---|---|---|---|---|---|---|---|----|-----------|
| Action 1 | ○ | × | ○ | ○ | × | ○ | × | ○ | × | ○ | ○ |
| Action 2 | ○ | × | × | × | ○ | × | × | ○ | × | × | × |
| Action 3 | ○ | × | ○ | ○ | × | ○ | ○ | × | ○ | × | × |
| Action 4 | ○ | × | × | × | ○ | ○ | ○ | ○ | × | ○ | ○ |

safety. However, certain skiing movements entail inherent risks that may lead to the unavailability of some human body key points, as demonstrated in Fig. 7 (3).

## Application test

To verify the practicality of our proposed system, we collaborated with eleven highly experienced ski coaches to conduct a real-world evaluation of our dynamic ski motion capture and human posture detection model, which is based on wireless device tracking. In Table 2, ten professional skiing instructors executed four predetermined skiing maneuvers, which were then predicted and classified using our system to determine their conformity to standard or non-standard movements. The subjective ratings of these ten ski instructors were employed to assess the four movements, with successful motion capture and human posture detection of skiing deemed accomplished if the scores aligned with the majority of instructors. It is worth noting that for each skiing maneuver, our system accurately identifies whether it adheres to standardization or not, which provides invaluable technical support for the dynamic tracking of skiing and human posture detection.

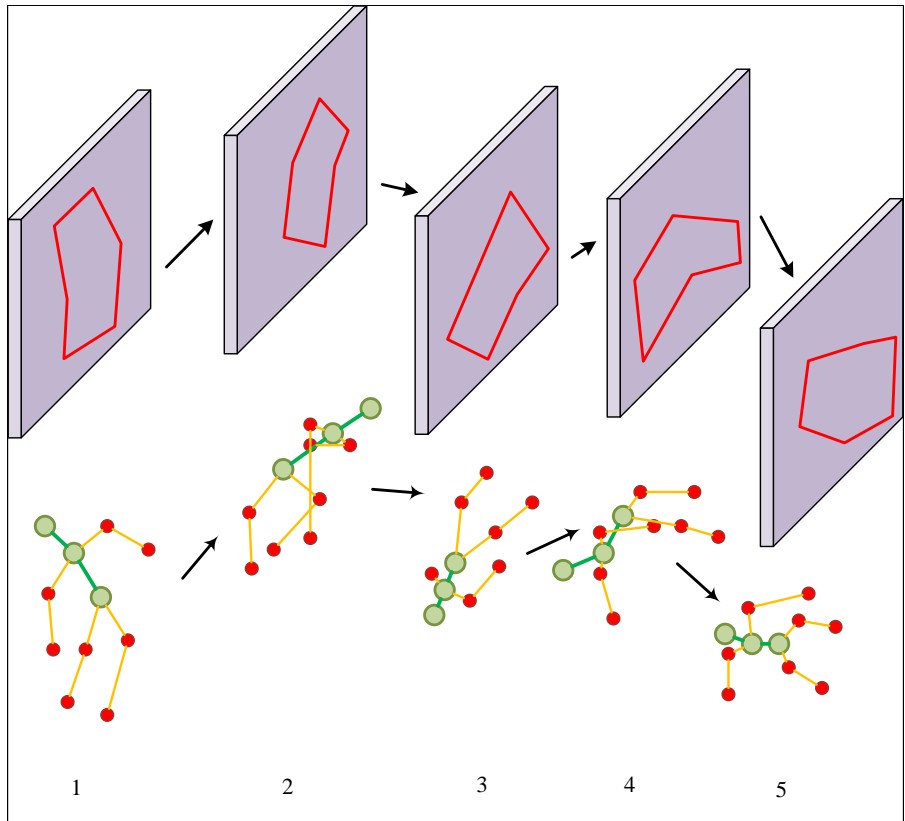

**Figure 7** **The training of our method and other methods.**

## CONCLUSION

In order to accomplish real-time monitoring and dynamic capture of skiing personnel postures, we introduced a dynamic skiing motion capture and human posture detection model that exploits a wireless device tracking. By establishing a service base station, we facilitated personnel location and tracking using wireless equipment. Additionally, we leveraged image information obtained by wireless devices to construct a human posture detection model that standardizes skiing movements. Our spatio-temporal Transformer structure empowered the dynamic capture of successive frames of human skiing. Our experimental outcomes demonstrated the high accuracy of our approach in real-time skiing personnel location, comprehensive human posture detection, and skiing dynamics analysis, thereby provided crucial technical support for ensuring skiing venue safety. Our experimental outcomes demonstrated the high accuracy of our approach in real-time skiing personnel location, comprehensive human posture detection, and skiing dynamics analysis, which can achieve the F value of 81.98%, thereby providing crucial technical support for ensuring skiing venue safety. In the future, we will explore to construct the more concise model to boost the sports supervision.

## ACKNOWLEDGEMENTS

We thank the anonymous reviewers whose comments and suggestions helped to improve the manuscript.

### Funding

This article was supported by the Humanities and Social Science Foundation of Ministry of Education, study on the supporting system of physical health and exercise promotion of rural residents under the Rural revitalization Strategy (No.21YJC890053). The funders had no role in study design, data collection and analysis, decision to publish, or preparation of the manuscript.

### Grant Disclosures

The following grant information was disclosed by the authors:
Humanities and Social Science Foundation of Ministry of Education, study on the supporting system of Physical health and exercise promotion of rural residents under the Rural revitalization Strategy: No. 21YJC890053.

### Competing Interests

The authors declare there are no competing interests.

### Author Contributions

- Xiaohua Wu conceived and designed the experiments, prepared figures and/or tables, and approved the final draft.
- Jian Yi performed the experiments, authored or reviewed drafts of the article, and approved the final draft.
- Yongtao Shi analyzed the data, prepared figures and/or tables, and approved the final draft.
- Gong Zhang performed the computation work, authored or reviewed drafts of the article, and approved the final draft.

### Data Availability

The data is available at Ski 2DPose Dataset: https://www.epfl.ch/labs/cvlab/data/ski-2dpose-dataset/.

### Supplemental Information

Supplemental information for this article can be found online at http://dx.doi.org/10.7717/peerj-cs.1618#supplemental-information.

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
