# Peer review of "Design of dynamic ski motion capture and human posture detection model based on spatio-temporal Transformer and wireless device tracking technology"

_PeerJ Computer Science, doi:10.7717/peerj-cs.1618_

## Round 0.1 · original submission · Major Revisions

Please address all the changes carefully and resubmit the updated version of the paper.
Please also improve the language of the paper.

Reviewer 1 ·

Basic reporting

Aiming at the importance of posture requirements in skiing events, this paper proposes a model of ski motion capture and human posture detection, and finally achieves model convergence and completes training objectives more effectively. Although the author has achieved some achievements, there are still many areas in this paper that need to be improved:

Experimental design

The author should make a paragraphed statement of the contribution;
There are just descriptions of other papers but not a real analysis applied to this proposed method. What is the originality of this research?
The illustrations and contents in the paper do not match, such as Figure 1 in line 133, I suggest the author to check carefully and correct;
IV. The author uses the focus structure to transform the feature of the input image. Please explain the reason for choosing this method;
V. The underlying network architecture includes CSPNet and FPN architectures. What are the advantages of using them to extract features?
VI. Add “wireless device tracking technology” to the title;
VII. The innovation of the method proposed by the author is not strong, please highlight the innovation of the model;
VIII. What are the advantages of the method proposed by the author to capture ski action dynamically?
IX. Recent references should be added that it is better to include some references from good journal papers.

Validity of the findings

validations have been performed but can be further improved as per suggestions above.

Additional comments

I have no additional comments

·

Basic reporting

Skiing, as a widely popular sport, necessitates the real-time maintenance of correct posture during movement. This paper present a dynamic skiing motion capture and human posture detection model that leverages wireless device tracking. Although the method model in this paper has certain effects, it still has the following shortcomings:
1. There are too few keywords to summarize the main idea, I suggest reorganizing paragraphs to add keywords;
2. Why use Mosica data enhancement to process the data set in this paper ?
3. How does the proposed model balance sensitivity and feature preservation compared to LSTM and Transformer?
4. The parameters of formula (1) and formula (2) are not explained;
5. How does the confidence branch update template add to the model and enhance the robustness of the algorithm? I don't really understand, please elaborate;
6. Combined with the content of the paper, please explain why the wireless device is used to confirm the location;
7. The conclusion is too short and lacks an introduction of practical value and future expectations;
8. Authors should pay more attention to the introduction of data sets, and if necessary, should add the process or method of data preprocessing, which will help other scholars to reference and learn from this study;
9. The author needs to go over the text carefully and clarify the figures and tables.

Experimental design

See above

Validity of the findings

See above

---

## Round 0.2 · Major Revisions

The original Academic Editor is not available and so I am taking over handling this submission.

Please address the comments from Reviewer 3 and revise the manuscript accordingly.

Reviewer 1 ·

Basic reporting

The authors have incorporated my suggestions , therefore, i feel that it could be accepted

Experimental design

The resutls are valid and seems to be okay to me

Validity of the findings

Validity are satisfactor

Additional comments

i have no more comments.

·

Basic reporting

I accept the manuscript

Experimental design

I accept the manuscript

Validity of the findings

I accept the manuscript

Additional comments

I accept the manuscript

Reviewer 3 ·

Basic reporting

1. The author must revise and clarify the figures and tables.

Experimental design

1. Is image resolution affect your model performance? What is the image resolution when getting from the wireless device tracking? And what is the optimal image resolution of your model for both accuracy and computer efficiency? The author should add a detail about the image resolution used in this research
2. The author should add the comparison of model performance before and after using Mosica data augmentation.
3. Why do demands lead to choosing each frame split from 0.3 to 10 seconds? What second of each frame split author use in the application test and why?
4. The result said the author’s method ranked first in precision but Table 1 shows that the Blip with 80.32% is the highest. I suggest the author check carefully and correct it.
5. Since this is a real-time issue and the author's model requires a long inference time, model processing times should be measured and compared to other models.

Validity of the findings

No comment

Additional comments

No comment

---

## Round 0.3 · Minor Revisions

Please revise Figure 6 as suggested by the reviewer.

Reviewer 3 ·

Basic reporting

The background provides adequate information to viewers. Each figure and table is structured and presented correctly. The result and the hypothesis are compatible. The language used is clear and straightforward. All terms and concepts were defined precisely.

Experimental design

The research corresponds with the journal's Aims and Scope. The findings clearly support and clarify the study's request. The approaches are described sufficiently in detail. The source code is made publically available.

Validity of the findings

The results of the experiments validate the findings. The conclusions are detailed and informative.

Additional comments

The authors have addressed all of my concerns in this revised version. I am satisfied with this version and encourage its publication in this journal, despite the need to check for adding information into figures like:
+ Adding unit information on the vertical and horizontal axes in Figure 6

---

## Round 0.4 · accepted · Accept

The authors have addressed all the comments from the reviewers. The manuscript can be accepted for publication.